# Virtual Coriolis-Force-Based Mode-Matching Micromachine-Optimized Tuning Fork Gyroscope without a Quadrature-Nulling Loop

**DOI:** 10.3390/mi14091704

**Published:** 2023-08-31

**Authors:** Yixuan Wu, Weizheng Yuan, Yanjun Xue, Honglong Chang, Qiang Shen

**Affiliations:** 1School of Mechanical Engineering, Northwestern Polytechnical University, Xi’an 710072, Chinayuanwz@nwpu.edu.cn (W.Y.);; 2MOE Key Laboratory of Micro and Nano Systems for Aerospace, Northwestern Polytechnical University, Xi’an 710072, China

**Keywords:** MEMS gyroscope, mode-matching, matching error, virtual Coriolis force

## Abstract

A VCF-based mode-matching micromachine-optimized tuning fork gyroscope is proposed to not only maximize the scale factor of the device, but also avoid use of an additional quadrature-nulling loop to prevent structure complexity, pick-up electrode occupation, and coupling with a mode-matching loop. In detail, a mode-matching, closed-loop system without a quadrature-nulling loop is established, and the corresponding convergence and matching error are quantitatively analyzed. The optimal straight beam of the gyro structure is then modeled to significantly reduce the quadrature coupling. The test results show that the frequency split is narrowed from 20 Hz to 0.014 Hz. The scale factor is improved 20.6 times and the bias instability (BI) is suppressed 3.28 times. The observed matching accuracy demonstrates that a mode matching system without a quadrature suppression loop is feasible and that the proposed device represents a competitive design for a mode-matching gyroscope.

## 1. Introduction

A gyroscope is a device for detecting the rotation rate of its carrier, and is one of the most important sensors for applications in fields such as navigation and industrial control [1]. MEMS-based gyroscopes are small and can be batch-fabricated at a low cost [2] compared with other gyroscope technologies such as photonic gyroscopes [3], nuclear magnetic gyroscopes [4], or fiber-optic gyroscopes [5]. However, the measurement accuracy of MEMS gyroscopes is limited due to machining errors. To improve the precision of MEMS gyroscopes, numerous approaches have been used, including the amelioration of the fabrication process [6], control of the vibration mode [7], temperature compensation [8], and back-end signal processing [9]. To benefit from the mode matching technology, the performance of MEMS gyroscopes has significantly improved to achieve navigation-grade accuracy [6]. This technology mainly relies on the fact that Coriolis-induced displacement in the sense mode reaches a maximum when the frequency split between the drive and sense modes of the gyroscope is zero, which means that the scale factor of the gyroscope is optimal.

The common mode matching scheme can be divided into three approaches. The first is the structure topology design of the gyroscope to achieve inherent mode matching. A ring gyroscope with L-shaped spokes based on a non-uniform radial width was designed to suppress the frequency split from 180 Hz to 50 Hz [10,11]. An elliptic-shaped structure connecting a thick ring alternately with multiple thin rings achieved the mode matching of 43 ppm at a center frequency of 69 kHz [12,13]. The frequency split of a micro-machined gyroscope was reduced to sub-Hz values with a temperature stability of 0.05 Hz over a 130 °C temperature range by placing critical mechanical elements, including coupling springs, anchors, and suspensions, at the center of the resonator structure [14]. 

The second approach to reduce the frequency split is employing a fabrication trimming scheme to modify the mass distribution via selective polysilicon deposition, laser trimming, or a focus-ion process. For example, a focus-ion beam process was used to trim the geometry to achieve the mode matching of 0.02 Hz at a resonant frequency of about 6 kHz [15]. A quantitative mass deposition process guided by a modification equation for the ring structure was devised to reach a mode matching level of 0.08 Hz [16]. Femtosecond laser trimming was applied with a mass-stiffness decoupling design to narrow the frequency split to 0.4 Hz [17].

The last widely used approach is an electrostatic tuning scheme based on the electrostatic spring softening technology to achieve mode matching by tuning the resonant frequency of the sense mode with a feedback DC voltage, which is a function of the characteristic signal reflecting the frequency split. For this category of method, the key issue is how to extract the related signal to the frequency split. Currently, existing methods for extracting the frequency split can be divided into three types, including (1) offline calibration, (2) signal symmetry comparison, and (3) small-signal-induced phase relation analysis.

For the offline calibration method, according to the previously defined quantitative relations between the frequency split, tuning DC voltage and angular rate output based on an offline experimental test, a lookup table (LUT) was established to identify the appropriate tuning voltage during the actual running to achieve mode matching. The LUT was built to regulate the frequency split of 0.01 Hz and to achieve an overall output bias drift of 0.76°/s over a 90 °C range [18]. A three-layer back propagation neural network controller was constructed to achieve a frequency split of 0.3 Hz over the temperature range from −40 °C to 80 °C by fitting the tuning voltage beforehand with the temperature [19]. Mode matching based on this method can be achieved quickly. Nevertheless, it is not easy to deal with the in-run, long-term mode matching uncertainty induced by mechanical fatigue deformation, residual stress change, or unexpected environment disturbances, such as vibration and shock.

With respect to the signal symmetry comparison method, double-side signal characteristics, including amplitude symmetry or noise power spectrum symmetry, are usually extracted as the matching error reference to judge the degree of mode matching. The symmetry of the both-sides band power spectrum of thermal noise in the sense mode was calculated to establish the functional relation between the degree of mode matching and the symmetrical error, achieving the frequency splits of 1.5 Hz [20,21] and 0.28 Hz [22], respectively. For high-quality gyroscopes, the noise characteristics dominated by the interface circuit are susceptible to environment disturbances, such as space electromagnetic coupling and stray capacitances fluctuation, which leads to mode matching accuracy deterioration. Two amplitude values of the signals at the lower and upper sides of the resonant frequency were compared by design-tuning the control loop to detect an amplitude difference, which will theoretically converge to zero when the two modes are matched. Nevertheless, due to the intrinsic, double-sided and non-uniform characteristic of high-Q gyroscopes, the slight asymmetry response was not fully eliminated [23].

With respect to the phase relation methods, the quadrature error is a typical characteristic excitation signal for mode matching [24,25,26]. In this way, the phase difference of the quadrature signal in the pick-up signal of the sense mode, with respect to the displacement of the drive mode, is extracted to generate a control error signal for the tuning voltage, where a 90° phase difference constituting a 0 V control error indicates the realization of mode matching. Nevertheless, the response to the angular rate input can couple into the control error and lead to a matching error when the sensor is of working status. 

Another improved phase relation method is developed based on the virtual Coriolis force (VCF) [27,28]. In this way, an external signal for simulating the Coriolis force with the corresponding frequency and phase information was loaded to special electrodes in the sense mode to excite the vibration of the sense mode. An induced pick-up signal was demodulated by the reference signal in the phase with Coriolis force to generate the control error signal, which trended to zero when the mode matching was reached. Since the VCF method is based on the closed-loop classical control theory, it is a real-time mode matching method, which avoids the calibration for each sensor to significantly reduce experimental complexity. More importantly, the phase relation between the VCF and the pick-up signal is only decided by the matching degree of the internal gyroscope, and it is not affected by external disturbances. In reported works, the VCF method is used to narrow the frequency split to 0.03 Hz and sub-0.1 Hz on the tuning fork gyroscope (TFG) [29] and disk resonator gyroscope (DRG) [30], respectively. 

Nevertheless, according to the precondition of the effectiveness of this VCF method, the quadrature error of the sensor can be suppressed as much as possible, which is usually realized through an additional closed quadrature-nulling loop [29,30,31];otherwise, the residual quadrature error, referring to the frequency split information, is an output to deteriorate the phase relation and reduce the sensitivity improvement of the gyroscope [28]. However, the additional quadrature suppression loop results in the complex gyroscope control circuit and reduction in the scale factor due to the occupation of sense pick-up electrodes. More importantly, the feedback of a quadrature-nulling loop may be coupled with the feedback of the mode-matching loop, leading to the leakage of stiffness trimming from the axis of quadrature stiffness to the axis of the sense mode. For example, with respect to the tuning fork gyroscope (TFG), the coexistence of slide-film excitation and squeeze-film excitation is why the electrostatic force is generated two-dimensionally and further simultaneously trims the stiffness of quadrature coupling and the sense mode [23]. With respect to the disk resonator gyroscope (DRG), the coupling is caused by the interaction between the sense mode stiffness and the quadrature coupling stiffness, due to the continuous distribution of the stiffness of the rings [32].

Considering the drawbacks of the phase relation method based on an additional quadrature suppression loop for the circuit complexity, occupation of electrodes, and coupling between quadrature nulling and mode matching, a simplified VCF-based mode-matching gyroscope without a quadrature-nulling loop and with an optimized TFG structure to achieve a sufficient structural quadrature suppression for mode matching is presented in this work. The proposed gyroscope without a quadrature-nulling loop has advantages in two aspects. On the one hand, as the quadrature-nulling loop is removed, the electrodes used for quadrature calibration can be used for detecting the sense mode displacement, and thus, the scale factor is maximized. On the other hand, the absence of the quadrature feedback means that the stiffness axis of the sense mode is only acted on by the frequency tuning voltage, and thus, the mode matching can be maintained more stably.

This work is organized as follows. The corresponding matching error caused by the quadrature error is theoretically analyzed in Section 2. Then, the optimization of the TFG structure is proposed and the effect of quadrature suppression is simulated. Third, the designed mode matching system with the estimated parameters of the gyroscope is simulated.

## 2. Theoretical Analysis of VCF-Based Mode Matching and Matching Error

### 2.1. Mode Matching System Design

Figure 1 shows the whole VCF-based mode matching schematic consisting of the mechanical part and electrical part. In the mechanical part, TFG is the object of the presented scheme, as shown in Figure 1b. In detail, TFG mainly consists of a drive, couple, and sense frames, which are marked by blue, green, and red, respectively. Due to the widely known tuning fork topology, the description of the Coriolis effect principle of TFG is omitted here. The electrical part contains three modules. The first module is the vibration-stabilizing closed-loop circuit for the drive mode, which is marked in blue. The second one is the equivalent electrical signal of the induced virtual Coriolis force, which is marked in red. The last one is the presented mode-matching circuit in the sense mode, which is marked in orange. In the mode-matching circuit, the total force, $F_{tot}$, consisting of Coriolis force, $F_{c}$, virtual Coriolis force, $F_{vc}$, and quadrature force, $F_{q}$, drives the sense mode to move. This displacement of sense mode y is then used as a voltage signal, $V_{sp}$, by the interface circuit. Then, the $V_{sp}$ is divided into circuit units for angular an velocity output and a mode-matching, closed loop. In the mode-matching loop, $V_{sp}$ is demodulated by the in-phase demodulation signal, $V_{dm}$, to obtain $V_{sdm}$ as a function of the phase shift, $\varphi_{s}^{d}$, between $F_{tot}$ and y. Through a low-pass filter, a comparator, and a proportional-integral (PI) controller, DC tuning voltage, $V_{t}$, is obtained and fed into the frequency tuning electrode of the sense mode. Based on the electrostatic softening effect, the resonant frequency of the sense mode, $\omega_{s}$, is tuned by $V_{t}$ and continuously varied until it equals to the resonant frequency of the drive mode $\omega_{d}$. The detailed deduction is given as follows.

First, $F_{c}$, $F_{vc}$, and $F_{q}$, which are components of $F_{tot}$, are defined in Equation (1). $F_{c}$ is the mechanical Coriolis force induced by the Coriolis effect under the angular rate input, $\Omega_{in}$, with a frequency of $\omega_{in}$. The amplitude of Coriolis force, $F_{c}$, is proportional to that of the angular rage input, $\Omega_{IN}$. $F_{vc}$ is the electrostatic force produced by the VCF excitation voltage, $V_{vc}$, which is obtained by amplifying the pick-up signal of the drive mode, $V_{dp}$, with the fixed gain, $K_{vc}$. According to the Coriolis effect theory and the principle of charge sensitive amplifier (CSA), all $F_{c}$, $V_{dp}$, $V_{vc}$, and $F_{vc}$ have in-phase relations. Thus, $F_{vc}$ is considered as artificial Coriolis force caused by the DC ($\omega_{in}$= 0) angular rate input, and it is defined as the VCF. The amplitudes of $F_{C}$, $F_{VC}$, and $F_{Q}$ are considered as fix values due to the closed-loop amplitude control in the drive mode.
(1)$$\left\{\begin{array}{l} F_{c}=F_{C}sin\left(\omega_{d}t\right)cos\left(\omega_{in}t\right)\\ F_{vc}=F_{VC}sin\left(\omega_{d}t\right)\\ F_{q}=F_{Q}cos\left(\omega_{d}t\right) \end{array}\right.$$

Then, the transfer function from $F_{tot}$ to the pick-up signal $V_{sp}$ is expressed as follows:(2)$$H_{s}\left(s\right)=\frac{L[V_{sp}(t)]}{L[F_{tot}(t)]}=\frac{K_{sp}/m_{s}}{s^{2}+\frac{\omega_{s}}{Q_{s}}s+\omega_{s}^{2}}$$
where $K_{sp}$ is the gain from y to $V_{sp}$, $m_{s}$ is the mass of the sense mode, $\omega_{s}$ is the resonant frequency of the sense mode, and $Q_{s}$ is the quality factor of the sense mode. According to the theory of the second-order system, $V_{sp}$ in Equation (2) can be solved as follows:(3)$$\left\{\begin{array}{cc} V_{sp}= & \begin{array}{c} \underset{response\;to\;F_{c}}{\underbrace{\frac{F_{C}}{2}\;[G_{s}^{d+in}sin\left(\omega_{d+in}t+\varphi_{s}^{d+in}\right)+G_{s}^{d-in}sin\left(\omega_{d-in}t+\varphi_{s}^{d-in}\right)]}} \end{array}\\ & +\underset{response\;to\;F_{vc}}{\underbrace{F_{VC}G_{s}^{d}sin\left(\omega_{d}t+\varphi_{s}^{d}\right)}}+\underset{response\;to\;F_{q}}{\underbrace{F_{Q}G_{s}^{d}sin\left(\omega_{d}t+\varphi_{s}^{d}+\frac{\pi }{2}\right)}}\\ \omega_{d\pm in} & =\omega_{d}\pm \omega_{in} \end{array}\right.$$
(4)$$\left\{\begin{array}{l} G_{s}^{d}=\frac{K_{sp}}{m_{s}\sqrt{\frac{\omega_{d}^{2}{\left(\Delta \omega +\omega_{d}\right)}^{2}}{Q_{s}^{2}}+{\left[{\left(\Delta \omega +\omega_{d}\right)}^{2}-\omega_{d}^{2}\right]}^{2}}}\\ \varphi_{s}^{d}=-arctan\left[\frac{\omega_{d}(\Delta \omega +\omega_{d})/Q_{s}}{{\left(\Delta \omega +\omega_{d}\right)}^{2}-\omega_{d}^{2}}\right] \end{array}\right.$$
(5)$$\left\{\begin{array}{l} G_{s}^{d+in}=\frac{K_{sp}}{m_{s}\sqrt{\frac{\omega_{d+in}^{2}{\left(\Delta \omega +\omega_{d}\right)}^{2}}{Q_{s}^{2}}+{\left[{\left(\Delta \omega +\omega_{d}\right)}^{2}-\omega_{d+in}^{2}\right]}^{2}}}\\ \varphi_{s}^{d+in}=-arctan\left[\frac{\omega_{d+in}(\Delta \omega +\omega_{d})/Q_{s}}{{\left(\Delta \omega +\omega_{d}\right)}^{2}-\omega_{d+in}^{2}}\right] \end{array}\right.$$
(6)$$\left\{\begin{array}{l} G_{s}^{d-in}=\frac{K_{sp}}{m_{s}\sqrt{\frac{\omega_{d-in}^{2}{\left(\Delta \omega +\omega_{d}\right)}^{2}}{Q_{s}^{2}}+{\left[{\left(\Delta \omega +\omega_{d}\right)}^{2}-\omega_{d-in}^{2}\right]}^{2}}}\\ \varphi_{s}^{d-in}=-arctan\left[\frac{\omega_{d-in}(\Delta \omega +\omega_{d})/Q_{s}}{{\left(\Delta \omega +\omega_{d}\right)}^{2}-\omega_{d-in}^{2}}\right] \end{array}\right.$$

In Equation (3), $V_{sp}$ comprises three terms, including $F_{c}$, $F_{vc}$, and $F_{q}$. In detail, the gain $G_{s}$ and the phase shift $\varphi_{s}$ from $F_{tot}$ to $V_{sp}$ can be deduced through the functions about frequency split Δω, where $\Delta \omega =\omega_{s}-\omega_{d}$. In the mode-matching loop, in order to extract the voltage amplitude, $V_{sp}$ is demodulated by $V_{dm}=V_{DM}sin(\omega_{d}t)$, which is derived from $V_{dp}$ multiplied by the fixed gain $K_{dm}$. $V_{sdm}$ is deduced as follows:(7)$$\begin{array}{cc} V_{sdm}(\varphi_{s}^{d})= & V_{sp}V_{dm}\\ = & \frac{V_{DM}}{2}\{\frac{F_{C}G_{s}^{d+in}}{2}\left[cos\left(\omega_{in}t+\varphi_{s}^{d+in}\right)-cos\left(2\omega_{d}t+\omega_{in}t+\varphi_{s}^{d+in}\right)\right]\\ & +\frac{F_{C}G_{s}^{d-in}}{2}\left[cos\left(\omega_{in}t-\varphi_{s}^{d-in}\right)-cos\left(2\omega_{d}t-\omega_{in}t+\varphi_{s}^{d-in}\right)\right]\\ & +\frac{F_{VC}G_{s}^{d}}{2}\left[cos\left(\varphi_{s}^{d}\right)-cos\left(2\omega_{d}t+\varphi_{s}^{d}\right)\right]\\ & +\frac{F_{Q}G_{s}^{d}}{2}\left[-sin\left(\varphi_{s}^{d}\right)+sin\left(2\omega_{d}t+\varphi_{s}^{d}\right)\right]\} \end{array}$$

From Equation (7), it can be seen that $V_{sdm}$ contains the signals of three classes of components distinguished by frequency: the DC signals about $\varphi_{s}^{d}$, the low-frequency signals with frequency of $\omega_{in}$, and the double-frequency signals with frequency of $2\omega_{d}$. Then, a low-pass filter (LPF) is designed after $V_{sdm}$ reaching the DC component $V_{comp}$:(8)$$V_{comp}(\varphi_{s}^{d})=\frac{V_{DM}G_{s}^{d}}{2}[\left(F_{C}+F_{VC}\right)cos\left(\varphi_{s}^{d}\right)-F_{Q}sin\left(\varphi_{s}^{d}\right)]$$

Herein, $V_{comp}$ is only related to the phase shift $\varphi_{s}^{d}$. Subsequently, a subtracter with negative input, $V_{comp}$, and a positive input, $V_{ref}=0$, are designed to obtain the error signal $V_{err}$:(9)$$\begin{array}{ll} V_{err}\left(\varphi_{s}^{d}\right) & =V_{ref}{-V}_{comp}\\ & =-\frac{V_{DM}G_{s}^{d}}{2}[\left(F_{C}+F_{VC}\right)cos\left(\varphi_{s}^{d}\right)-F_{Q}sin\left(\varphi_{s}^{d}\right)] \end{array}$$

$V_{err}$ is then input into the PI controller to pursue the DC tuning voltage, $V_{t}$, which is fed into the frequency-tuning electrode of the sense mode. According to electrostatic spring softening theory [33,34], the stiffness variation of the sense mode $k_{t}$ versus tuning voltage $V_{t}$ can be expressed as follows:(10)$$\left\{\begin{array}{l} k_{t}(V_{t})=\varepsilon_{0}HL\left(\frac{1}{D_{1}^{3}}+\frac{1}{D_{2}^{3}}\right)[N_{t}{\left(V_{p}-V_{t}\right)}^{2}+(N_{sp}+N_{vc}){V_{p}}^{2}]\\ \omega_{s}(V_{t})=\sqrt{\frac{k_{s0}-k_{t}}{m_{s}}} \end{array}\right.$$

The design parameters of the sensor structure in Equation (10) are defined in Table 1. 

The process reflected by Equations (3)−(10) evolves continuously until $V_{err}$ equals to zero, which is proved by Equations (11)−(14) in the following Section 2.2. In this case, if $F_{Q}$ is suppressed and can be neglected, $\varphi_{s}^{d}$ satisfies the relation $\varphi_{s}^{d}=\pi /2$ according to Equation (9). Furthermore, according to Equation (4), it is deduced that Δω=0, suggesting that mode matching is realized. In practice, $F_{Q}$ is not strictly controlled and it produces matching errors within the system. Thus, the influence of $F_{Q}$ is analyzed in Section 2.2.

### 2.2. Quantification of Matching Errors

As mentioned above, the designed system is self-stabilizing and requires quadrature suppression for mode matching. However, the quadrature force is not strictly controlled with the absence of the quadrature-suppression circuit in this system for maximizing the detection sensitivity of $V_{sp}$. Herein, the matching error caused by $F_{Q}$ is mathematically analyzed.

As the mode matching system is designed with stability conditions, it can be linearized around its stable point to study the matching error in a stable state. In this case, the equivalent system concerning how Δω is established, as shown in Figure 2, and the nonlinear relations between two nets are linearized. For example, the square root relation between $k_{s}$ and $\omega_{s}$ is replaced by the fixed gain, $G_{\omega }$.

By taking $V_{err}$, $V_{t}$, and Δω as the output variables, three closed-loop transfer functions are constructed:(11)$$\left\{\begin{array}{l} \left\{\left[L\left(V_{err}\right)\left(K_{P}+\frac{K_{I}}{s}\right)G_{k}+\frac{k_{s0}}{s}\right]G_{\omega }-\frac{\omega_{d}}{s}\right\}G_{\varphi }F_{TOT}G_{F}+\frac{F_{Q}G_{Q}}{s}=L\left(V_{err}\right)\\ \left\{\left[\left(L\left(V_{t}\right)G_{k}+\frac{k_{s0}}{s}\right)G_{\omega }-\frac{\omega_{d}}{s}\right]G_{\varphi }F_{TOT}G_{F}+\frac{F_{Q}G_{Q}}{s}\right\}\left(K_{P}+\frac{K_{I}}{s}\right)=L\left(V_{t}\right)\\ \left[\left(L\left(\Delta \omega \right)G_{\varphi }F_{TOT}G_{F}+\frac{F_{Q}G_{Q}}{s}\right)\left(K_{P}+\frac{K_{I}}{s}\right)G_{k}+\frac{k_{s0}}{s}\right]G_{\omega }-\frac{\omega_{d}}{s}=L\left(\Delta \omega \right) \end{array}\right.$$
where $F_{TOT}=F_{C}+F_{VC}+F_{Q}$ and $L\left(*\right)$ is the Laplace operator. By solving Equation (11), the Laplace transformations of $V_{err}$, $V_{t}$, and Δω are obtained as follows:(12)$$\left\{\begin{array}{l} L\left(V_{err}\right)=-\frac{G_{Q}F_{Q}+G_{\varphi }G_{F}F_{TOT}\left(G_{\omega }k_{s0}-\omega_{d}\right)}{\left(G_{k}G_{\omega }G_{\varphi }G_{F}K_{P}F_{TOT}-1\right)s+G_{k}G_{\omega }G_{\varphi }G_{F}K_{I}F_{TOT}}\\ L\left(V_{t}\right)=-\frac{\left(K_{I}+K_{P}s\right)\left[G_{Q}F_{Q}+G_{\varphi }G_{F}F_{TOT}\left(G_{\omega }k_{s0}-\omega_{d}\right)\right]}{\left(G_{k}G_{\omega }G_{\varphi }G_{F}K_{P}F_{TOT}-1\right)s^{2}+G_{k}G_{\omega }G_{\varphi }G_{F}K_{I}F_{TOT}s}\\ L\left(\Delta \omega \right)=-\frac{G_{k}G_{\omega }G_{Q}K_{I}F_{Q}+s\left(G_{k}G_{\omega }G_{Q}K_{P}F_{Q}+G_{\omega }k_{s0}-\omega_{d}\right)}{\left(G_{k}G_{\omega }G_{\varphi }G_{F}K_{P}F_{TOT}-1\right)s^{2}+G_{k}G_{\omega }G_{\varphi }G_{F}K_{I}F_{TOT}s} \end{array}\right.$$

By conducting an inverse Laplace transformation for Equation (12), the solutions in the time domain of $V_{err}$, $V_{t}$, and Δω are solved as follows:(13)$$\left\{\begin{array}{l} V_{err}=-\frac{G_{Q}F_{Q}+G_{\varphi }G_{F}F_{TOT}\left(G_{\omega }k_{s0}-\omega_{d}\right)}{G_{k}G_{\omega }G_{\varphi }G_{F}K_{P}F_{TOT}-1}e^{-\frac{1}{T}t}\\ V_{t}=-\frac{G_{Q}F_{Q}+G_{\varphi }G_{F}F_{TOT}\left(G_{\omega }k_{s0}-\omega_{d}\right)}{G_{k}G_{\omega }G_{\varphi }G_{F}F_{TOT}}-\frac{G_{Q}F_{Q}+G_{\varphi }G_{F}F_{TOT}\left(G_{\omega }k_{s0}-\omega_{d}\right)}{G_{k}G_{\omega }G_{\varphi }G_{F}F_{TOT}\left(G_{k}G_{\omega }G_{\varphi }G_{F}K_{P}F_{TOT}-1\right)}e^{-\frac{1}{T}t}\\ \Delta \omega =-\frac{G_{Q}F_{Q}}{G_{\varphi }G_{F}F_{TOT}}+\left[\frac{G_{Q}F_{Q}}{G_{\varphi }G_{F}F_{TOT}}-\frac{G_{k}G_{\omega }G_{Q}K_{P}F_{Q}+G_{\omega }k_{s0}-\omega_{d}}{G_{k}G_{\omega }G_{\varphi }G_{F}K_{P}F_{TOT}-1}\right]e^{-\frac{1}{T}t} \end{array}\right.$$
where $1/T=G_{k}G_{\omega }G_{\varphi }G_{F}K_{I}F_{TOT}/\left(G_{k}G_{\omega }G_{\varphi }G_{F}K_{P}F_{TOT}-1\right)$ is the damping coefficient. With increasing the time, t, the exponential terms in Equation (13) converge to zero and the stable points of $V_{err}$, $V_{t}$, and Δω converge to the following equations:(14)$$\left\{\begin{array}{l} V_{err}=0\\ V_{t}=-\frac{G_{Q}F_{Q}}{G_{k}G_{\omega }G_{\varphi }G_{F}F_{TOT}}-\frac{G_{\omega }k_{s0}-\omega_{d}}{G_{k}G_{\omega }}\\ \Delta \omega =-\frac{G_{Q}F_{Q}}{G_{\varphi }G_{F}F_{TOT}} \end{array}\right.$$

Obviously, the stable point of $V_{err}$ is always equal to zero. This feature is provided by the principle of the PI controller and not affected by $F_{Q}$. In contrast, the stable point of Δω is a residual term about $F_{Q}$, which causes the matching error of the system. Equation (14) indicates $V_{err}=0$ when the system is stable, and the phase shift, $\varphi_{s}^{d}$, can be deduced using Equation (9):(15)$$\varphi_{s}^{d}=arctan\left(\frac{F_{C}+F_{VC}}{F_{Q}}\right)$$

By combining Equations (4) and (15), the stable solution of Δω is expressed as follows:(16)$$\Delta \omega =\left\{\frac{F_{Q}}{2Q_{s}\left(F_{C}+F_{VC}\right)}+\sqrt{{\left[\frac{F_{Q}}{2Q_{s}\left(F_{C}+F_{VC}\right)}\right]}^{2}+1}-1\right\}\omega_{d}$$

According to Equation (16), the matching error is affected by $F_{C}+F_{VC}$ and $F_{Q}$. Specifically, the matching error is proportional to the ratio of $F_{Q}$ to $F_{C}+F_{VC}$ when it satisfies Equation (17):(17)$${\left[\frac{F_{Q}}{2Q_{s}\left(F_{C}+F_{VC}\right)}\right]}^{2}\ll 1$$

Thus, $\left|F_{C}+F_{VC}\right|$ should be reduced as much as possible with respect to $\left|F_{Q}\right|$ for the matching error reduction. On the contrary, the increase in $\left|F_{Q}\right|$ will enlarge the matching error. In addition, according to Figure 2, $F_{C}+F_{VC}$ constitutes the loop gain of the mode matching system and its polarity should be constant in working to keep the negative feedback characteristics of the mode matching system; otherwise, if the value of $F_{C}+F_{VC}$ exceeds the zero point, the system can possibly oscillate. In summary, the design of the mode matching system turns to an optimization problem:(18)$$\left\{\begin{array}{l} max\;\left|F_{C}+F_{VC}\right|\\ s.t.\;F_{C}+F_{VC}>0,\;F_{VC}>0 \end{array}\right.$$

In this system, $F_{VC}$ is designed positively, with a fixed value when the drive loop of gyro is stable. Then, the value of $F_{C}+F_{VC}$ should be discussed in two cases:

Case $F_{C}\geq 0$

The case that $F_{C}\geq 0$ corresponds to the situation where the positive direction angular rate is the input. In this case, $F_{C}+F_{VC}$ always satisfies the constraint condition in Equation (18) and it is increased with increasing the angular rate. Meanwhile, the matching error is decreased by the input angular rate, according to Equation (16).

Case $F_{C}<0$

The case that $F_{C}<0$ corresponds to the situation when the negative direction angular rate is the input. In this case, we have $F_{C}+F_{VC}=F_{VC}-\left|F_{C}\right|$. To guarantee the constraint condition in Equation (18), it is required that $\left|F_{C}\right|<F_{VC}$. Combined with Equation (16), it is revealed that the increment in the angular rate in the negative direction magnifies the matching error, and $F_{VC}$ is the boundary value of the angular rate in negative direction.

The numerical analysis of the matching error with the designed range of $F_{C}$, $F_{VC}$, and $F_{Q}$ is proposed. To explicitly compare $F_{C}$, $F_{VC}$, and $F_{Q}$, they are equivalently transformed into the form of amplitudes of angular rate $\Omega_{IN}$, $\Omega_{VC}$, and $\Omega_{Q}$, respectively.

First, $\Omega_{IN}$ is the test condition provided by the rate table, and thus it is known as the setting value. In this paper, it varies from −49°/s to −50°/s.

Then, the value of $F_{VC}$ is selected because it is the constant excitation of the in-run mode matching system. $F_{VC}$ is calculated according to the electrostatic effect of the mechanical capacitor.
(19)$$F_{VC}=\varepsilon_{0}N_{vc}HL\left(\frac{1}{D_{1}^{2}}-\frac{1}{D_{2}^{2}}\right)V_{p}V_{VC}$$
where $V_{VC}$ is the amplitude of $V_{vc}$. Meanwhile, $F_{VC}$ is also related to $\Omega_{VC}$ under the Coriolis effect.
(20)$$F_{VC}=2\Omega_{VC}m_{s}\dot{x}$$
where $\dot{x}=\omega_{d}x$ is the velocity of drive mode and x is the displacement of the drive mode. Combining Equations (19) and (20), the virtual angular rated input $\Omega_{VC}$ is expressed by Equation (21), as follows:(21)$$\Omega_{VC}=\frac{\varepsilon_{0}N_{vc}HL}{2m_{s}\dot{x}}\left(\frac{1}{D_{1}^{2}}-\frac{1}{D_{2}^{2}}\right)V_{p}V_{VC}$$

In this paper, $\Omega_{VC}$ is 50°/s via trimming $V_{VC}$.

Third, $\Omega_{Q}$ is evaluated by the scale factor of the gyro. Considering that $\Omega_{in}$ is set as a DC rotation, it is obvious that
(22)$$\frac{\Omega_{Q}}{\Omega_{IN}}=\frac{V_{Q}}{V_{IN}}$$
where $V_{Q}$ and $V_{IN}$ are the DC voltages induced by $F_{Q}$ and $F_{C}$ at the output of the sense mode detection circuit, respectively. As $V_{IN}/\Omega_{IN}$ is the scale factor of the gyro, $\Omega_{Q}$ is finally equal to the ratio of $V_{Q}$ to the scale factor. Experimentally, $\Omega_{Q}$ can be estimated under 15°/s for the proposed gyro. Furthermore, the numerical relations between $\Omega_{IN}$, $\Omega_{Q}$, and Δf is calculated through Equation (16), where $F_{C},$ $F_{VC}$, and $F_{Q}$ are replaced by $\Omega_{IN}$, $\Omega_{VC}$, and $\Omega_{Q}$, respectively. Additionally, Δf=Δω/2π represents the matching error in unit of Hertz. Calculation results are shown in Figure 3. Figure 3a shows the continuous numerical relationship between Δf and $\Omega_{Q}$ with several samples of $F_{C}$, and Figure 3b corresponds to the opposite case. It should be noted that Δf increases significantly when $\Omega_{IN}$ approaches the boundary negative direction (−$\Omega_{VC}$). Thus, the merge of a negative measurement is required for the specific matching error target in operation. By comparing the effects of $\Omega_{Q}$ and $\Omega_{IN}$ on Δf, the target of Δf is set as 0.5 Hz for the negative measurement limit $\Omega_{IN}$ = −35°/s when $\Omega_{Q}$ = 15°/s, which is an empirical value in our previous design of TFG [35]. The negative measurement limit can be possibly expanded to −45°/s when $\Omega_{Q}$ is further suppressed below 5°/s, which is the estimated value referring to [36].

## 3. System Implementation

### 3.1. Optimization of the Gyro Sensor

The optimized TFG structure compared with the original prototype is demonstrated in Figure 4. The folded sense suspension beams and the sense-decoupled beams are replaced by the straight ones, which provide better stiffness along the drive axis of the gyro, thereby improving the oscillation stability of a couple frame and the rigid constraints of a sense frame along the drive axis. Consequently, the quadrature couple between the drive and sense modes is further suppressed.

The effect of quadrature suppression of TFG structures is verified through a mechanical simulation. The excitation for quadrature error is operated as shown in Figure 5. To produce the mode coupling, the deformation of the element mesh is used to drive the suspension beams for simulating the machining error, which induces the quadrature coupling between the drive and sense mode (Figure 5a). To produce the quadrature force, the drive mode is excited by adding a harmonic disturbance load to the drive excitation electrodes (Figure 5b). The frequency sweep of the harmonic disturbance is executed in the simulation and the displacement of the sense mode is recorded to evaluate the level of quadrature error. 

The simulation results are shown in Figure 6 and Figure 7. Figure 6 shows the distortion of the sense suspension beams caused by the quadrature coupling when the harmonic disturbance load is used for the drive excitation electrodes. By comparison, a significant distortion emerges on the sense suspension beams of the original structure, whereas the sense suspension beams of the optimized structure present a better resistance against the drive coupling. Figure 7 shows the frequency sweep of the displacement corresponding to the two structures of the quadrature coupling. The maximum amplitudes of displacement of the two structures are 0.53 um and 0.11 um, respectively. The quadrature errors $\Omega_{Q}$ of the two structures are equivalent to 17°/s and 3.5°/s, respectively.

### 3.2. Simulation System Construction

According to the designed system, a parameterized system model for the real circuit implementation is constructed as shown in Figure 8. The simulation parameters are summarized in Table 2. The simulation works consist of a transient response, stable matching error, and scale factor.

The transient response of the mode matching system varies relatively to $\Omega_{IN}$, as shown in Figure 9. The frequency split converges to zero during the full scale, from –45°/s to 50°/s, in the absence of the quadrature error. The setting time is about 0.6 s when $\Omega_{IN\;}$= 50°/s, whereas it is increased to about 11 s when $\Omega_{IN}$ gradually varies to −45°/s. This variation is caused by the closed-loop gain related to $\Omega_{IN}+\Omega_{VC}$ when the quadrature error is neglected according to the equivalent system in Figure 2.

The matching error caused by the quadrature error is simulated with a series of $\Omega_{IN}$ and $\Omega_{Q}$. The results are summarized in Table 3. The relations between Δf, $\Omega_{IN}$, and $\Omega_{Q}$ agrees well with the description in Section 2.2. The matching error is controlled to under 0.41 Hz within the measurement range of $\Omega_{IN}\;$> −45°/s when $\Omega_{Q}$ is suppressed under 5°/s.

The scale factor simulation is performed as shown in Figure 10. The discrete points are the simulated samples, and the solid lines are the corresponding first-order fitting lines. The fitted scale factors of the open-loop operation and closed-loop operation with the quadrature error varying from 0°/s to 15°/s are calculated as 0.983 mV/°/s, 16.095 mV/°/s, 16.082 mV/°/s, 16.089 mV/°/s, and 16.102 mV/°/s, respectively. The scale factor of the gyro with the mode matching system is improved 16.36 times on average, compared with that of the open-loop operation. Compared with the result in the case of $\Omega_{Q}$ = 0, the worst deterioration of the scale factor caused by the quadrature error is 0.12%. The nonlinearity of the open-loop operation and closed-loop operation with quadrature error varying from 0°/s to 15°/s is calculated as 0.056%, 0.042%, 0.079%, 0.084%, and 0.031%, respectively. The correlation between the nonlinearity and quadrature error cannot be observed through simulation. It can be considered that the proposed system without circuits for the quadrature suppression is feasible and acceptable with respect to both the scale factor improvement and the nonlinearity maintenance.

## 4. Experiment

### 4.1. TFG Sensor Fabrication

The fabrication of TFG sensor is divided into three parts: the through-silicon-via (TSV) process, the device process, and the wafer-level-package (WLP) process, as shown in Figure 11. The TSV process contains the following treatments: (a) etching the back side with TMAH; (b) deep reactive ion etching (DRIE) for trench; (c) filling trench with thermal oxide and in situ doped polysilicon; and (d) etching the cavity and removing the oxide layer. The device process contains the following treatments: (e) etching the shallow-bottom cavity and thermal oxidation; (f) vacuum bonding, grinding, and chemical mechanical polishing (CMP) the device layer; and (g) DRIE device wafer to release movable structures. The WLP process contains the following treatments: (h) vacuum-bonding the TSV wafer to the device wafer; (i) plasma-enhanced chemical vapor deposition (PECVD) of TEOS and reactive ion etching (RIE); (j) sputtering and dry-etching the metal layer; and (k) PECVD of TEOS and RIE.

### 4.2. TFG Sensor Tests

The implemented test board and TFG sensor are shown in Figure 12. The experiments include the open-loop sweep for the mode characteristics, the open-loop frequency tuning for verifying the electrostatic tuning effect, the transient response of drive mode for the demodulation signals, the start-up progress of the mode-matching circuit for evaluating the frequency split, and the scale factor, as well as the Allan variance for evaluating the overall performance.

The sweep tests of the drive and sense modes were conducted under the open-loop operation. The magnitude–frequency response and the phase–frequency response of the two modes are shown in Figure 13. It can be found that the resonant frequencies of the drive and sense modes are 6960.09 Hz and 6940.28 Hz, respectively. Additionally, the quality factors of the two modes are calculated with the 3dB bandwidth, which are $Q_{d\;}$= 24,857 and $Q_{s}\;$= 7977, respectively.

The open-loop frequency tuning test was conducted as shown in Figure 14. The resonant frequency of the drive mode is independent of the tuning voltage because of the fully decoupled structure. The resonant frequency of sense mode is firstly increased and then decreased when $V_{t}$ exceeds 5 V. This is because an initial stiffness softening is induced by the bias voltage, $V_{p}$, and is then offset by the improvement in $V_{t}$ according to Equation (10).

The transient response of the drive mode under the mode mismatching operation is shown in Figure 15. The drive mode proceeds about 3 s to reach a stable status. The drive excitation signal, $V_{de}$, is maintained at about 128 mV and the drive pickup signal $V_{dp}$ is controlled at 1 V. The drive mode oscillates at its resonant frequency of 6960.3 Hz, which is indicated by the antiphase between $V_{de}$ and $V_{dp}$. Furthermore, $V_{dp}$ is reused as the in-phase demodulation signal $V_{dm}$ and shifted with 90° to get the orthogonal demodulation signal $V_{dm-90}$, as shown in Figure 16.

The start-up progress of the mode-matching circuit and the oscillation of the sense mode are shown in Figure 17. In Figure 17a, the error voltage, $V_{err}$, and the tuning voltage, $V_{t}$, converge to 0 V and 715 mV after 2 s, respectively. The detected angular rate output voltage is 350 mV, which reflects the virtual angular rated input, $\Omega_{VC}$. The zoomed waveform in Figure 17b shows the phase difference of 88.18° between $V_{dp}$ and $V_{sp}$, which is close to the theory value of 90° when the gyro is under the mode matching operation. According to the relationship between the phase shift, $\varphi_{s}^{d}$, and frequency split, Δf, as expressed by Equation (4), Δf is controlled at about 0.014 Hz. It has been proven that without the additional circuits for quadrature suppression, the structure-optimized gyro sensor still provides an excellent performance of mode decoupling, and the residual quadrature error barely produces any effect on the mode matching.

The scale factors of both the mode matching and mismatching operations are tested as shown in Figure 18. The discrete points are the tested samples, and the solid lines are the corresponding first-order fitting lines within the range from −50°/s to 40°/s. The scale factor of the mode matching operation is calculated as 6.70 mV/°/s, which is improved 20.6 times compared with that of the mode mismatching operation (0.33 mV/°/s). This improvement in experimental scale factor conforms well to the estimated value in the corresponding system simulation. The nonlinearity of the mode matching operation is calculated as 1.77%, which is comparable to that of the mode mismatching operation of 1.61%. In addition, it is notable that the scale factor is deteriorated when $\Omega_{IN}$ exceeds 40°/s, where $\Omega_{VC}$ is nearly offset by $\Omega_{IN}$ and cannot provide enough gain for the mode-matching loop. Herein, the mode mismatching occurs and decreases the scale factor.

Finally, the Allan variances of both mode matching and mismatching operations are calculated as shown in Figure 19. As the sense mode of the gyro oscillates at its resonant frequency, the signal–noise ratio (SNR) is also improved. The data are acquired with the sampling rate of 10 Hz during 2 h. The bias instability (BI) of the mode matching operation is calculated as 9.60°/h, which is suppressed 3.25 times compared with that of the mode mismatching operation of 31.53°/h.

The comparison of the performance in different works is given in Table 4. The matching error of this work is at a similar level of [29,31], which indicates that the quadrature suppression circuit is unnecessary in a VCF-based mode-matching circuit relying on the proposed TFG decoupling design. For the specific numerical difference, the implementation of a prototype, such as the selection of operational amplifier or voltage reference, may be the primary influencing factor [29]. The BI suppression of this work is not outstanding for the lack of the test environment control. Notably, the work in [31] presents an extraordinary scale factor improvement and BI suppression. This feature probably benefits from a high-quality factor of 150,000 of the gyro. 

## 5. Conclusions

In this paper, a simplified VCF-based mode-matching micromachine-optimized TFG is proposed. For the gyroscope sensor, the structure is optimized to provide a better performance of structural quadrature suppression. The related theoretical analyses, simulations, and experiments are implemented. The overall performance of the tested prototype is that the frequency split is narrowed from 20 Hz to 0.014 Hz. The scale factor is improved from 0.33 mV/°/s to 6.70 mV/°/s. The BI is suppressed from 31.53°/h to 9.60°/h. The level of BI agrees well with the accuracy requirement of the tactical grade and can satisfy applications, such as those in industrial electronics and those that are low-end tactical [1]. Benefiting from the simplicity of the circuit design, this gyroscope is particularly suitable in applications requiring a low cost and low power consumption, such as unmanned aerial vehicles (UAVs), movable energy harvesting systems, and consumer electronics. Alongside that, the tested matching error represents an outstanding degree of mode matching, and the times of BI suppression are at a similar level to that in other works. These results effectively prove the feasibility of the proposed non-quadrature-nulling mode matching scheme for avoiding the structure complexity, pick-up electrode occupation, and coupling between the quadrature-nulling loop and the mode-matching loop. As a preview of the work, it is hoped that it this will be applied in higher-end fields, such as attitude and heading reference systems (AHRSs) with a finer interface circuit to reduce the BI.

## Figures and Tables

**Figure 1 micromachines-14-01704-f001:**
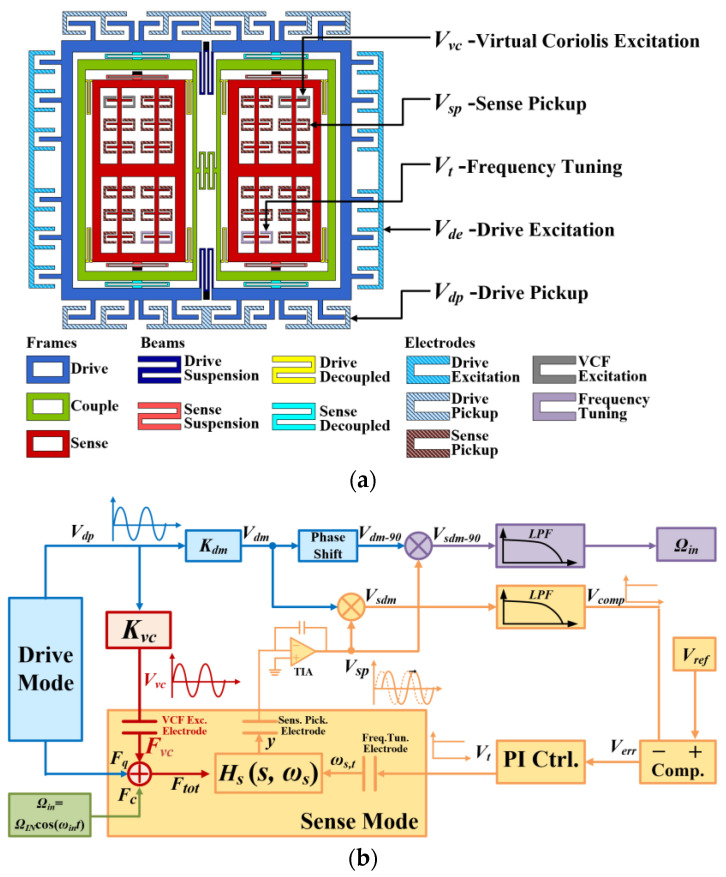
Schematic of (**a**) mode matching system with (**b**) the topology of the gyro sensor.

**Figure 2 micromachines-14-01704-f002:**
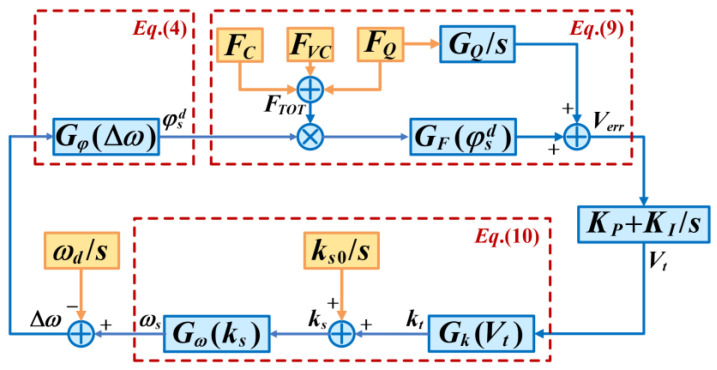
Schematic of the frequency split controlling.

**Figure 3 micromachines-14-01704-f003:**
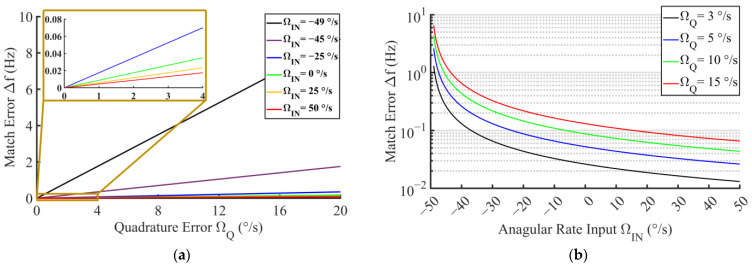
Matching error versus varying (**a**) quadrature error and (**b**) angular rate input.

**Figure 4 micromachines-14-01704-f004:**
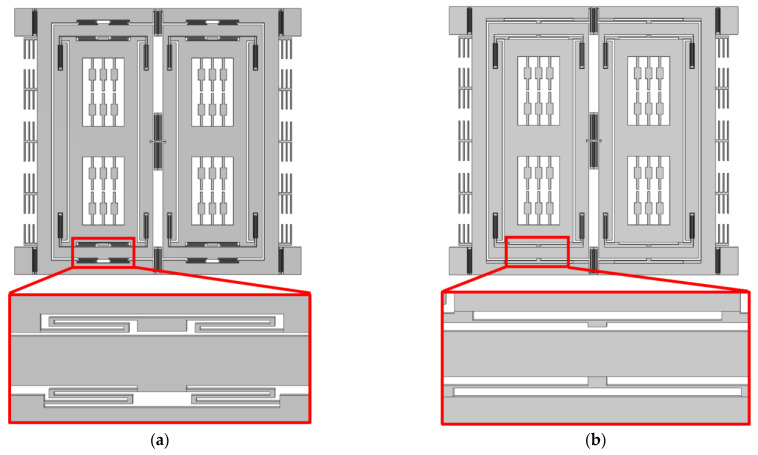
Structure comparison of (**a**) original structure and (**b**) optimized structure.

**Figure 5 micromachines-14-01704-f005:**
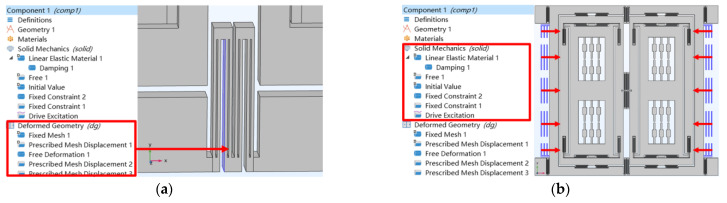
Steps of quadrature error simulation (**a**) mesh deformation and (**b**) load definition.

**Figure 6 micromachines-14-01704-f006:**
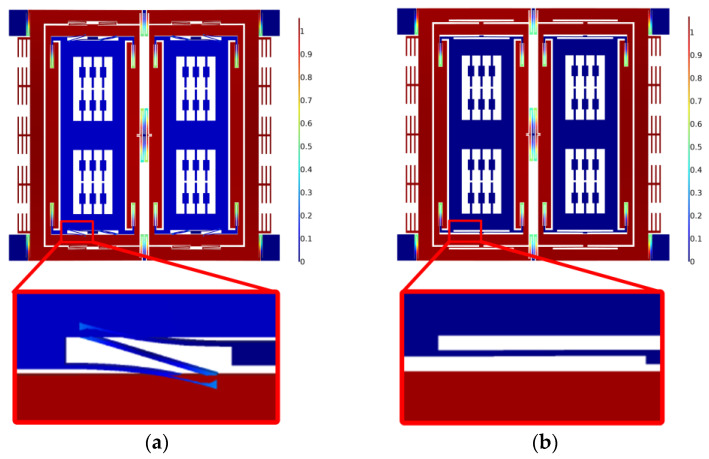
Quadrature couple simulation of (**a**) the original structure and (**b**) the optimized structure.

**Figure 7 micromachines-14-01704-f007:**
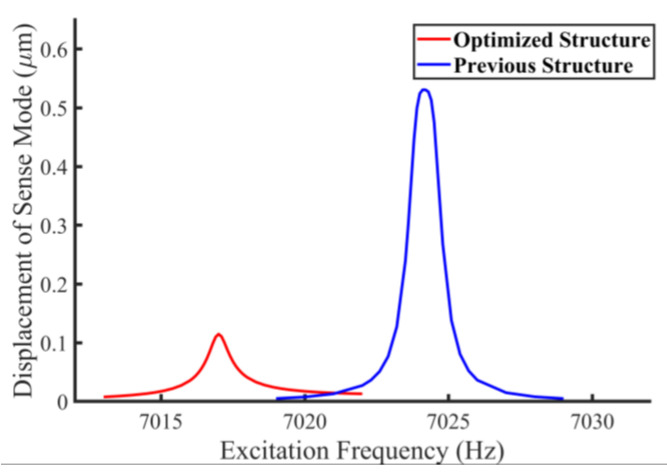
Comparison of coupled displacement in the sense mode.

**Figure 8 micromachines-14-01704-f008:**
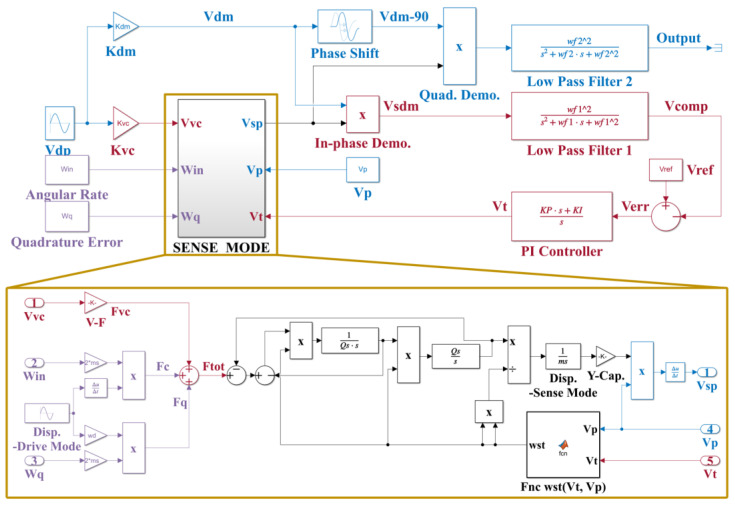
Schematic of the simulation model.

**Figure 9 micromachines-14-01704-f009:**
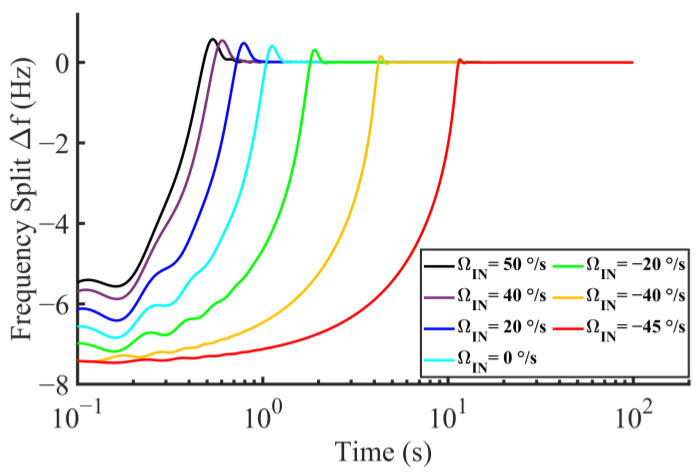
Transient response with different angular rate inputs.

**Figure 10 micromachines-14-01704-f010:**
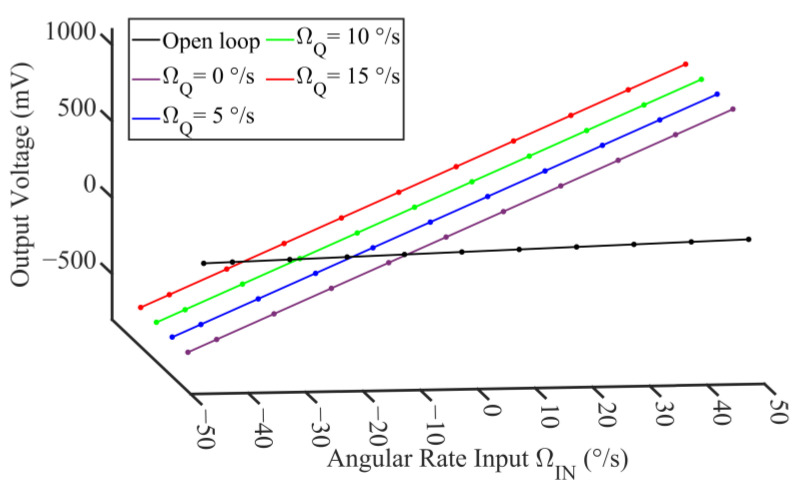
Scale factor simulation with different quadrature errors.

**Figure 11 micromachines-14-01704-f011:**
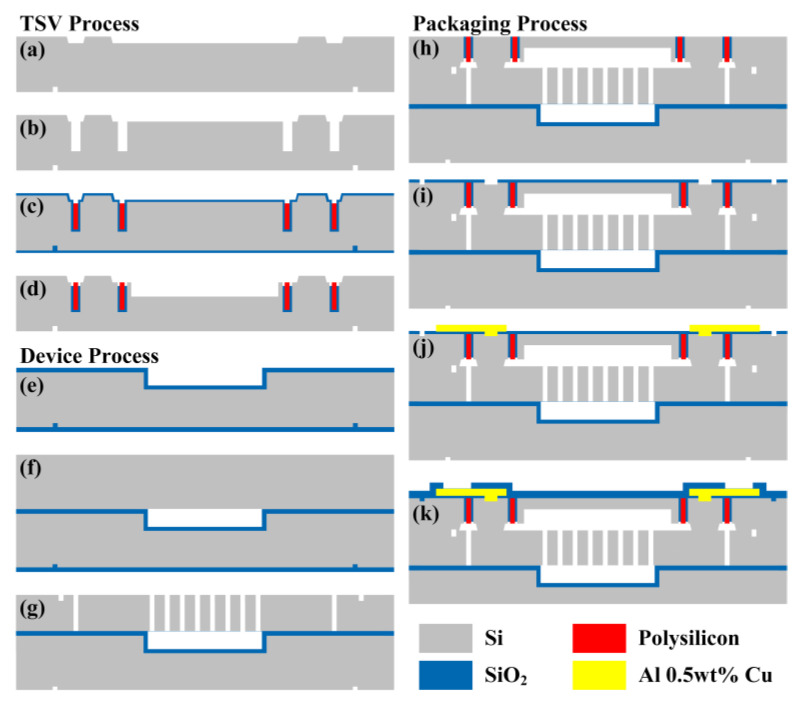
Fabrication process of TFG sensor: (**a**) etching the back side with TMAH; (**b**) deep reactive ion etching (DRIE) for trench; (**c**) filling trench with thermal oxide and in situ doped polysilicon; (**d**) etching the cavity and removing the oxide layer; (**e**) etching the shallow-bottom cavity and thermal oxidation, (**f**) vacuum bonding, grinding, and chemical mechanical polishing (CMP) the device layer; (**g**) DRIE device wafer to release movable structures; (**h**) vacuum-bonding the TSV wafer to the device wafer; (**i**) plasma-enhanced-chemical-vapor deposition (PECVD) of TEOS and reactive ion etching (RIE); (**j**) sputtering and dry-etching the metal layer; and (**k**) PECVD of TEOS and RIE.

**Figure 12 micromachines-14-01704-f012:**
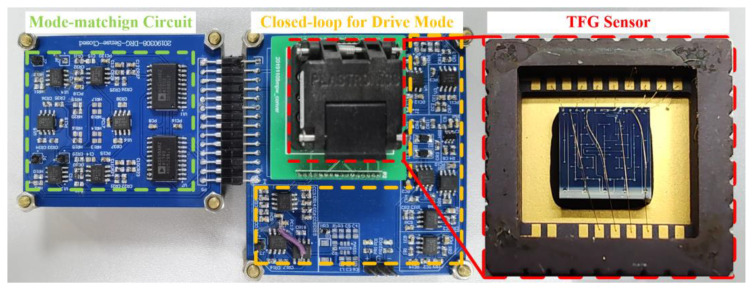
PCB test board with the TFG sensor.

**Figure 13 micromachines-14-01704-f013:**
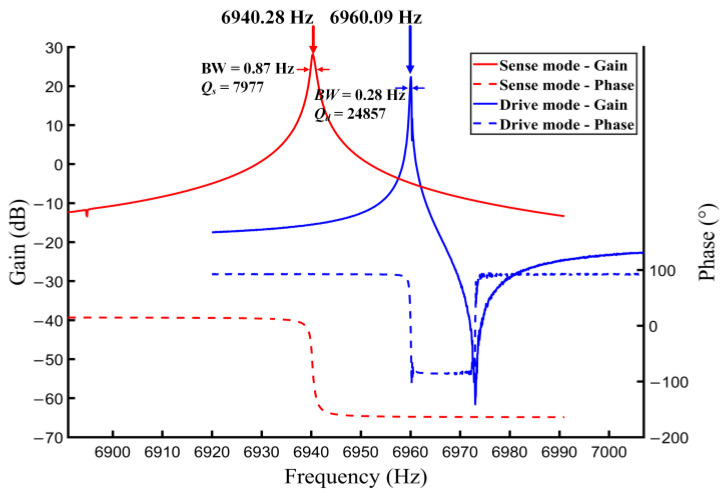
Sweep tests of the drive and sense modes under open-loop operation.

**Figure 14 micromachines-14-01704-f014:**
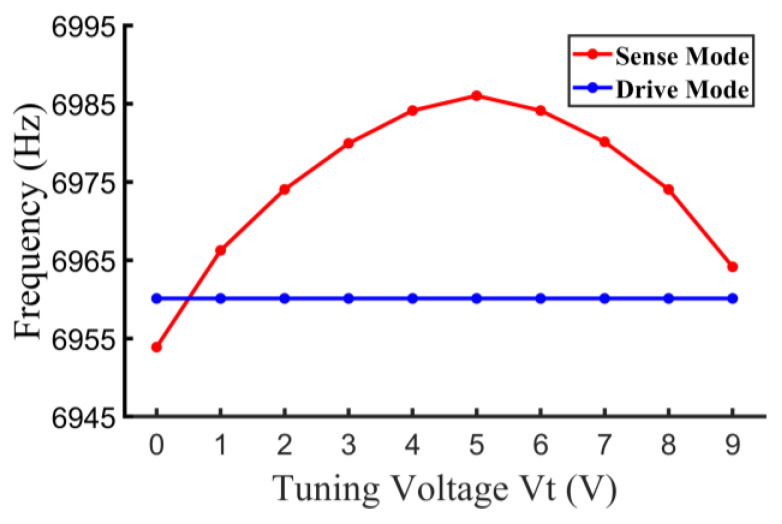
Open-loop frequency tuning test.

**Figure 15 micromachines-14-01704-f015:**
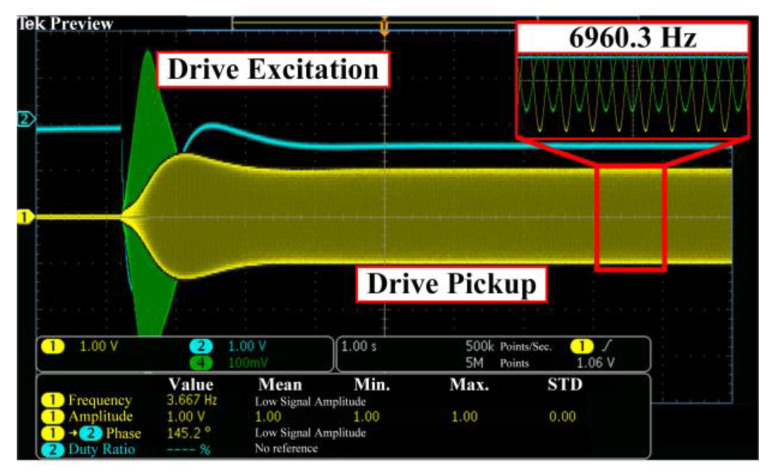
Transient response of the drive mode.

**Figure 16 micromachines-14-01704-f016:**
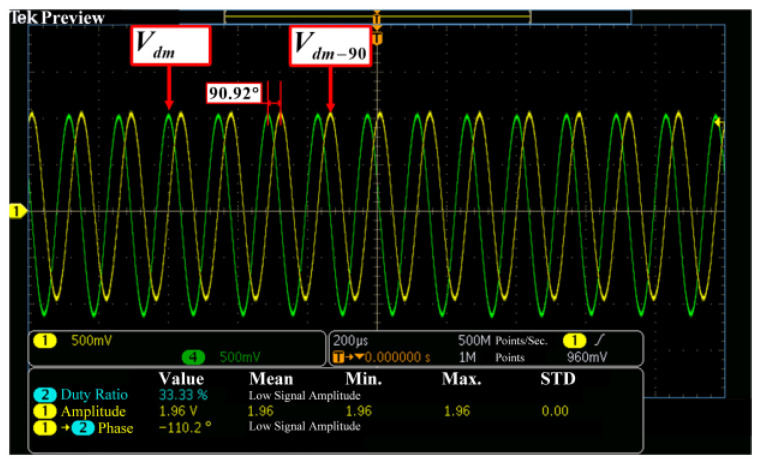
Phase shift of the demodulation signals.

**Figure 17 micromachines-14-01704-f017:**
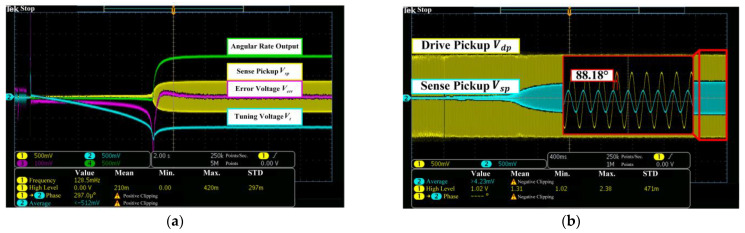
Start-up progress of (**a**) mode matching and (**b**) the sense mode.

**Figure 18 micromachines-14-01704-f018:**
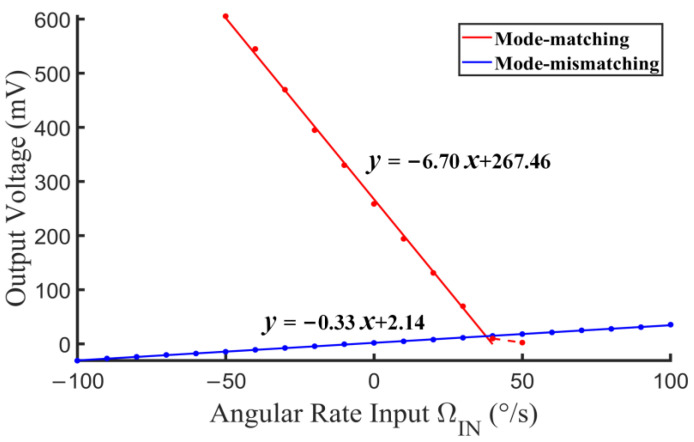
Scale factor test of mode matching and mismatching.

**Figure 19 micromachines-14-01704-f019:**
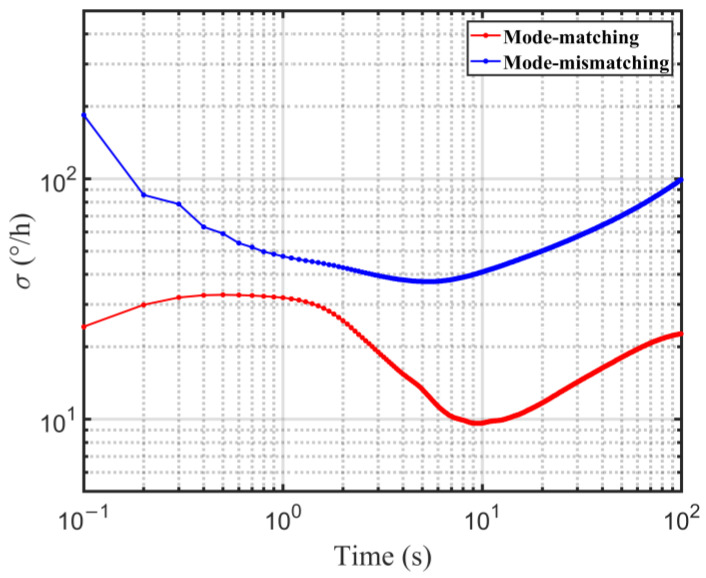
Allan variance test of mode matching and mismatching.

**Table 1 micromachines-14-01704-t001:** Parameters of the sensor structure.

Symbol	Description
$\varepsilon_{0}$	vacuum dielectric constant
$m_{s}$	mass of the sense mode
$k_{s}$	$k_{s}=\omega_{s}^{2}m_{s}$, stiffness of sense mode
$k_{s0}$	stiffness of the sense mode when $V_{t}=0$
H	thickness of the device layer
L	overlap length of the combs
$D_{1}$	upper gap of the electrodes
$D_{2}$	lower gap of the electrodes
$N_{sp}$	number of capacitor pairs for the sense pick-up
$N_{vc}$	number of capacitor pairs for the VCF feedback
$N_{t}$	number of capacitor pairs for frequency tuning

**Table 2 micromachines-14-01704-t002:** Simulation parameters.

Parameter	Value
$f_{d}$	7000 Hz
$f_{s}$	6990 Hz
$Q_{s}$	8000
$\Omega_{IN}$	−45°/s~50°/s
$\Omega_{VC}$	50°/s
$\Omega_{Q}$	5°/s~15°/s

**Table 3 micromachines-14-01704-t003:** Matching error (Hz) versus quadrature error and angular rate input.

		$\Omega_{IN}$(°/s)				
		−45	−20	0	20	50
$\Omega_{Q}$(°/s)	5	0.41	0.09	0.06	0.05	0.04
	10	0.88	0.16	0.10	0.08	0.08
	15	1.36	0.15	0.15	0.11	0.09

**Table 4 micromachines-14-01704-t004:** Summary of the performance of VCF-based works.

Ref.	Matching Error (Hz)	SF Improvement (Times)	BI Suppression (Times)
This work	0.014	20.6	3.28
[28]	0.6	5.60	-
[29]	0.03	16.8	-
[30]	0.1	17.9	10.9
[31]	0.005	68.8	23.1

## Data Availability

The data that support the findings of this study are available from the corresponding author upon reasonable request.

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
