# Peer review of "Virtual Coriolis-Force-Based Mode-Matching Micromachine-Optimized Tuning Fork Gyroscope without a Quadrature-Nulling Loop"

_micromachines, 2023, doi:10.3390/mi14091704_

Round 1

Reviewer 1 Report

This work studies a VCF-based mode-matching micromachined optimized tuning fork gyroscope. The author established a mode-matching closed-loop system without quadrature-nulling loop, and analyzed the corresponding convergence and matching error quantitatively.

However, there are several technical issues needed to be addressed before further consideration for publication in Micromachines Journal. Please address the following comments:

1.The format needs to be revised. The first letter of a sentence should be capitalized (line 15 in paper). Please check the whole manuscript and revise it.

2.The paper introduced how to use virtual Coriolis force(VCF) to realize modal matching. However, It is suggested that a detailed presentation of the sources of the VCF could be compiled to better help the reader understand the principles of the methodology.

3.The section 2.1 of the paper is narrated in the order from the mechanical part to the electrical part, but the order of the the (a) and (b) subgraphs in Figure 1 is reversed from the order of the paper. Please swap the order of the two subgraphs in Figure 1.

4.This paper did a lot of formula derivation work quantify the matching error In Section 2.2. It is mentioned in the paper that the numerical relation among Ω??, Ω? and ?? can be calculated according to Equation(16) (line 257-259 in paper), but the specific formula of conversion is not given clearly. Please give the appropriate derivation process.

The author can clearly explain the research content and results, but the language can be further revised and improved.

Reviewer 2 Report

The Authors propose a tuning-fork gyroscope. The reported results have been achieved by both simulation and experiments. The manuscript deserves the publication after addressing the following comments:

1. In the Introduction, an overview of competing technologies has to be reported (see, i.e., Planar photonic gyroscopes for satellite attitude control. In 2017 7th IEEE International Workshop on Advances in Sensors and Interfaces (IWASI) (pp. 167-169). IEEE, 2017; The nuclear magnetic resonance gyroscope: a review. The Journal of Navigation40(3), 366-384, 1987; The development of micro-gyroscope technology. Journal of Micromechanics and Microengineering19(11), 113001, 2009).

2. Since the achieved resolution strongly limits the potential applications, please report a survey of the tuning fork gyroscope applications.

Reviewer 3 Report

The work described in the paper is appreciable. A simplified VCF-based mode-matching gyroscope without the quadrature-nulling loop and leveraging the optimized TFG structure to realize a sufficient structural quadrature suppression for mode matching is presented. The paper may be accepted once its revision is done.

However, there are some issues need to be addressed which are given as below:

 1.The article carried out a lot of formulae derivation. However, there were some clerical errors in writing present (line 190 and 208-209 in paper). Please check and modify the whole manuscript.

 2.Section 2.2 of the paper concluded that the ratio of ?? and ??+??? has an effect on the matching error (line 222-224 in paper). However, the effect of FQ on matching error was not explicitly concluded. It is suggested that a concluding statement be added to echo the above (line 192 in paper) and to enrich the need for structural optimisation in the later section.

 3.The paper showed that ??, ???, and ?? can be equivalently converted to Ω??, Ω??, and Ω? (line 244-246 in paper). However, there were no explicit formulae derived to account for the conversion of Ω?? and Ω? correspondingly. It is recommended that this section be supplemented.

 4.No preview were given how the work will be applied in the future. It is suggested that some of the outstanding issues or future directions for exploration related to the research topic be pointed out. These questions can stimulate the reader's interest and provide ideas for follow-up research.
